# Piezo1 Regulation Involves Lipid Domains and the Cytoskeleton and Is Favored by the Stomatocyte–Discocyte–Echinocyte Transformation

**DOI:** 10.3390/biom14010051

**Published:** 2023-12-30

**Authors:** Amaury Stommen, Marine Ghodsi, Anne-Sophie Cloos, Louise Conrard, Andra C. Dumitru, Patrick Henriet, Christophe E. Pierreux, David Alsteens, Donatienne Tyteca

**Affiliations:** 1CELL Unit and PICT Platform, de Duve Institute, UCLouvain, 1200 Brussels, Belgium; amaury.stommen@uclouvain.be (A.S.); marine.ghodsi@uclouvain.be (M.G.); anne-sophie.cloos@uclouvain.be (A.-S.C.); patrick.henriet@uclouvain.be (P.H.); christophe.pierreux@uclouvain.be (C.E.P.); 2Center for Microscopy and Molecular Imaging (CMMI), Biopark Charleroi, Université Libre de Bruxelles, 6041 Gosselies, Belgium; louise.conrard@ulb.be; 3Louvain Institute of Biomolecular Science and Technology, UCLouvain, 1348 Louvain-la-Neuve, Belgiumdavid.alsteens@uclouvain.be (D.A.)

**Keywords:** Piezo1 distribution, Piezo1 chemical activation, Yoda1, membrane stiffness, erythrocyte morphology, cholesterol, GM1 ganglioside, spectrin cytoskeleton

## Abstract

Piezo1 is a mechanosensitive ion channel required for various biological processes, but its regulation remains poorly understood. Here, we used erythrocytes to address this question since they display Piezo1 clusters, a strong and dynamic cytoskeleton and three types of submicrometric lipid domains, respectively enriched in cholesterol, GM1 ganglioside/cholesterol and sphingomyelin/cholesterol. We revealed that Piezo1 clusters were present in both the rim and the dimple erythrocyte regions. Upon Piezo1 chemical activation by Yoda1, the Piezo1 cluster proportion mainly increased in the dimple area. This increase was accompanied by Ca^2+^ influx and a rise in echinocytes, in GM1/cholesterol-enriched domains in the dimple and in cholesterol-enriched domains in the rim. Conversely, the effects of Piezo1 activation were abrogated upon membrane cholesterol depletion. Furthermore, upon Piezo1-independent Ca^2+^ influx, the above changes were not observed. In healthy donors with a high echinocyte proportion, Ca^2+^ influx, lipid domains and Piezo1 fluorescence were high even at resting state, whereas the cytoskeleton membrane occupancy was lower. Accordingly, upon decreases in cytoskeleton membrane occupancy and stiffness in erythrocytes from patients with hereditary spherocytosis, Piezo1 fluorescence was increased. Altogether, we showed that Piezo1 was differentially controlled by lipid domains and the cytoskeleton and was favored by the stomatocyte–discocyte–echinocyte transformation.

## 1. Introduction

Piezo are mechanosensitive cation-permeable channels showing a preference for calcium (Ca^2+^) ions. This family regroups two proteins, Piezo1 and Piezo2, which are evolutionarily conserved and found in multiple eukaryotic organisms, organs and tissues [1,2]. Piezo proteins form large homotrimeric transmembrane (TM) structures presenting a ~40% homology but are not homologous to other ion channel families [3,4]. They present a propeller-like structure in which each protomer consists of 38 TM helices organized into 9 TM helical units of 4 TMs each, called the blades. The last 2 helices together with the extracellular cap and the C-terminal domain form the central pore that allows cation fluxes [5,6,7,8,9]. Physiologically, Piezo1 contributes to blood pressure, epithelial cell adhesion, blood vessel formation and development of vascular architecture, neutrophil recruitment, skeletal muscle remodeling and red blood cell (RBC) volume regulation [10,11,12,13,14,15]. Pathologically, Piezo1 has been shown to be deregulated in lymphatic dysplasia [16] and hereditary xerocytosis [17,18,19] but also in several cancers [20,21].

Two main paradigms are proposed for the regulation of Piezo1, the force-from-lipid and the force-from-filament models. In the force-from-lipid paradigm, Piezo1 sensitivity and/or cycle of activation are thought to be regulated by (i) specific lipids such as cholesterol (chol) and its association with TM proteins [22,23,24], phosphoinositides [25], as well as sphingomyelin (SM) and its ceramide product [26], (ii) plasma membrane (PM) transversal asymmetry through the flip-flop of phosphatidylserine (PS) [27] and (iii) PM tension, as illustrated by the importance of the ratio of saturated over unsaturated fatty acids [28] and the membrane dome hypothesis. The latter proposes that the highly curved membrane imposed by the Piezo1 shape contains enough energy to gate this protein upon tension-induced flattening [3,8]. In the force-from-filament model, Piezo1 regulation is thought to be regulated by internal (i.e., the cytoskeleton [29,30,31]) and/or external (i.e., extracellular matrix (ECM) [32,33]) tethers. Nowadays, a more cooperative model dependent on the cell type seems to emerge [34,35,36], highlighting the need of studying Piezo1 in appropriate cellular physiological contexts while considering both the membrane lipid and filament components.

RBCs represent an ideal model to address this question as Piezo1 is expressed and plays a role in Ca^2+^ fluxes at the RBC surface and both the cytoskeleton and the membrane contribute to RBC morphology and deformability, as detailed below. First, RBCs depend on the transient increase in intracellular Ca^2+^ levels for deformation [37], and Piezo1 is the main mechanosensitive channel physiologically expressed in those highly deformable cells [11,38]. We previously revealed that the intracellular Ca^2+^ content increases upon RBC chemical modulation by the specific allosteric modulator Yoda1 [39], confirming the implication of Piezo1 in Ca^2+^ influx (Appendix A). We also showed the presence of Piezo1 at the RBC surface and its heterogeneous distribution in submicrometric clusters of different sizes through both atomic force microscopy (AFM) and high-resolution confocal imaging [40]. Second, the RBC cytoskeleton is at the same time robust, dynamic and well-anchored to the PM thanks to two non-redundant anchorage complexes based on 4.1R and ankyrin proteins [12,41]. Third, the RBC membrane tension, which can be seen as the stretching–compression elastic stresses at every point along the membrane surface [42], is regulated by multiple partners including the PM lipid composition, the cytoskeleton membrane anchorage and the non-muscular myosin NMIIA [43,44]. Fourth, the RBC membrane presents a heterogeneous lateral distribution of lipids in the forms of both transient nanometric rafts enriched in sphingolipids and chol [45] and stable submicrometric domains. Those latter present differential composition, abundance, properties and roles in RBC deformation. Thus, domains mainly enriched in chol (named chol-enriched domains) are the most abundant, associate with both the RBC rim and dimple regions, exhibit the highest lipid order and gather in the high curvature areas upon RBC deformation. In contrast, domains co-enriched in SM and chol (named SM-enriched domains) or in GM1 ganglioside and chol (named GM1-enriched domains) associate almost exclusively with the RBC dimple and exhibit lower ordering than chol-enriched domains [46,47]. Moreover, several lines of evidence suggest that these two types of lipid domains contribute to Ca^2+^ exchange at the RBC surface. For instance, upon RBC deformation on stretchable PDMS chambers, the early and transient Ca^2+^ influx is concomitant with the increase in GM1-enriched domains, whereas the subsequent decrease in the Ca^2+^ levels is accompanied by the late increase in SM-enriched domains (Appendix A) [47]. In contrast, upon combination of the mechanical stimulus with the mechanosensitive channel inhibitor GsMTx4 peptide [48], all increases are abolished (Appendix A). Altogether, these data suggest that GM1-enriched domains might play a role in Piezo1 regulation, but this remains to be clearly demonstrated.

As mentioned above, RBCs present a typical discocyte shape characterized by rim and dimple regions resulting from a ~40% excess of PM surface area compared to a sphere of same volume [49,50]. The discocyte shape can be reversibly modulated toward stomatocytes and echinocytes or non-reversibly toward spherocytes [51,52,53]. The stomato–disco–echinocyte transformation can be explained through the bilayer couple hypothesis that claims that the two PM leaflets respond differently to perturbations but remain coupled to each other. Indeed, expansion of the outer leaflet leads to a convex form and, thus, echinocytes, while expansion of the inner leaflet leads to a concave form and, thus, stomatocytes. Those expansions are necessary and sufficient to explain shape changes [54,55].

We here assessed the potential contribution of the PM biomechanical properties, the PM lipid distribution and the cytoskeleton to Piezo1 regulation in relation with RBC morphology. To do so, we visualized Piezo1 and determined in parallel PM lipid domain abundance and distribution, transversal asymmetry, stiffness, membrane:cytoskeleton anchorage and morphology of healthy RBCs, modulated or not for chol content, and of cytoskeleton-defective RBCs. The specific modulator Yoda1 was used to induce Ca^2+^ influx through Piezo1 [39], while phorbol esters were used to activate protein kinase C (PKC) and thereby stimulate Ca^2+^ influx though the Cav2.1 channel [56].

## 2. Materials and Methods

### 2.1. Blood Collection and RBC Preparation

This study was approved by the Medical Ethics Committee of the Université catholique de Louvain, Brussels, Belgium. All donors gave written informed consent. Four healthy adult volunteers (1 man, C1, and 3 women, C2–C4) aged between 24 and 30 years were included in the study. C1 was arbitrarily taken as the reference condition. Four patients with hereditary spherocytosis (3 men and 1 woman aged between 16 and 74 years), named P1, P17, P18 and P19, respectively, by reference to our previous study [57], were also included. Blood was collected by venipuncture into K+/EDTA-coated tubes (except when otherwise stated), which were stored at 4 °C and used for a maximum of 4 days after collection. Before experiments, RBCs were isolated from other blood components by a 1:10 dilution in Dulbecco’s modified Eagle medium (DMEM containing 4.5 g/L glucose, 25 mM HEPES and no phenol red, Invitrogen, Waltham, MA, USA), washed twice by centrifugation at 200× *g* for 2 min and resuspended. This procedure allowed us to efficiently separate RBCs from platelets and white blood cells, as revealed by the absence of contamination in RBC preparations in our routine fluorescence imaging experiments [47].

### 2.2. RBC Chemical Treatments

All treatments were performed in suspension on washed isolated RBCs diluted 12 times in DMEM (~42,000 RBCs/µL). After each treatment, RBCs were centrifuged at 200× *g* for 2 min and resuspended in chemical agent-free DMEM, except when otherwise stated. For Piezo1 activation, RBCs were pre-conditioned at the appropriate temperature for 20 min and then incubated with the allosteric modulator Yoda1 (Biotechne, Minneapolis, MD, USA) for 30 s. Except when otherwise stated, the Yoda1 concentration was 50 nM for immunofluorescence experiments and 100 nM for all other experiments. To induce a PKC-mediated Ca^2+^ permeability [56], RBCs were incubated for 20 min at 37 °C with 3 µM phorbol 12-myristate 13-acetate (PMA; Sigma-Aldrich, Saint-Louis, MO, USA) and 30 nM of the protein phosphatase inhibitor calyculin A from *Discodermia calyx* (CalA; Sigma-Aldrich). These agents were maintained throughout the whole experiment. To deplete membrane chol, RBCs were incubated with the chol-removing agent methyl-β-cyclodextrin (mβCD; Sigma-Aldrich) for 15 min at 0.6 mM at 37 °C, except when otherwise stated. To replete membrane chol after depletion, RBCs were incubated with 7.5 µg/mL of mβCD:chol (Sigma-Aldrich) for 60 min and then in a mβCD:chol-free medium for an additional 90 min to allow time for newly inserted chol to distribute in the appropriate membrane regions.

### 2.3. RBC Morphology and Surface Areas Determination

The RBC morphology was evaluated on living RBCs in suspension in μ-dish IBIDI chambers. Images were captured with a Zeiss widefield fluorescence microscope (Observer.Z1; plan-Apochromat 63× or 100× 1.4 oil Ph3 objective). Only RBCs in a side view were analyzed to correctly distinguish the 4 RBC morphologies: discocytes, stomatocytes, echinocytes and spherocytes (Appendix A). For each condition, the proportion of each RBC population was determined and expressed as the percent of the total RBCs counted. Then, a delta of each morphology was applied between the experimental and the control conditions. The hemi-RBC surface area was determined on living RBCs spread on poly-L-lysine (PLL)-coated coverslips. The Fiji v.1.53 software was utilized, and the RBCs were manually surrounded (white dotted circles in Appendix A). RBC rim and dimple areas were determined on fixed RBCs dropped on PLL-coated coverslips with the Zen 3.5 software (Blue Edition, Zeiss). For this parameter, RBCs in the upper view were used as they allowed us to better distinguish the rim from the dimple. Practically, two areas were manually drawn: the whole RBC area determined by circling the outside of the RBC edge and the dimple area determined by circling the inner side of the RBC edge. The rim area was then obtained by subtracting the dimple region area from the whole one (Appendix A). Data were expressed as the percentage of the control condition or the percentage of dimple region area occupation in each experimental condition.

### 2.4. RBC Membrane Lipid Vital Imaging

SM and GM1 were visualized by insertion of the fluorescent lipid analogs BODIPY-SM (0.6 µM for 20 min at RT) or -GM1 (1 µM for 20 min at 37 °C) into the PM of RBCs spread onto PLL-coated coverslips, as in [47]. Endogenous chol was decorated by the mCherry-Theta toxin fragment (0.6–1 µM for 20 min at RT) on RBCs in suspension and then spread onto PLL-coated coverslips, as in [58]. All coverslips were then placed upside down in LabTek chambers (Thermo Fisher Scientific, Waltham, MA, USA) filled with DMEM and observed with a Zeiss widefield fluorescence microscope as above. For quantification, the total number of lipid domains was assessed by manual counting on fluorescence images (purple insets in Appendix A) and reported to the hemi-RBC projected area determined with the Fiji software (white dotted circles in Appendix A). In some experiments, the abundances of chol-enriched domains in the rim and in the dimple were determined separately (white arrows and arrowheads in Appendix A).

### 2.5. RBC Intracellular Calcium Content Measurement

Intracellular Ca^2+^ content was determined using the Fluo-4AM (acetoxy methyl ester) probe (Thermo Fisher Scientific). Briefly, RBCs were first incubated with 3 µM Fluo-4AM in a 1.8 mM Ca^2+^-containing and Fe^2+^/Mg^2+^-free homemade medium (to avoid interaction of the probe with other bivalent cations) for 60 min at RT or 37 °C, followed by a de-esterification step in a probe-free medium for 30 min at the appropriate temperature (allowing the probe to bind Ca^2+^ ions), all in suspension in the dark. Then, labeled RBCs were analyzed by fluorimetry (Glomax; Promega, Madison, WI, USA) on a dark plate, and the hemoglobin (Hb) content was measured by spectrophotometry to normalize the data as in [47,59]. The data are finally expressed as percentages of the control condition.

### 2.6. RBC Piezo1 and Spectrin Immunofluorescence

For immunofluorescence on RBCs in suspension, RBCs were fixed for 15 min in suspension with a mix of 4% paraformaldehyde (PFA; Sigma-Aldrich) and 0.05% glutaraldehyde (Sigma-Aldrich). After immobilization on PLL-coated coverslips, RBCs were permeabilized with 1% (*v*/*v*) Triton X-100 for 20 min and blocked with 3% (*w*/*v*) bovine serum albumin (BSA; Sigma-Aldrich) as in [40]. This permeabilization step did not modify the Piezo1 cluster number but allowed for a better visualization as they were more brilliant than without permeabilization [40]. RBCs were then incubated with rabbit polyclonal antibody to Piezo1 (ProteinTech, Rosemont, IL, USA, 15939-1-AP; 1:50 dilution) or mouse monoclonal antibody to α-β-spectrin (Sigma-Aldrich, S3396; 1:100 dilution) in 0.2% (*w*/*v*) BSA in a moist environment for 120 min. After washing 3 × 3 min with PBS, RBCs were incubated with the secondary goat anti-rabbit Alexa 488 (for Piezo1 alone), goat anti-mouse Alexa 488 (for spectrin alone) or Alexa 647 (for spectrin in combination with Piezo1) antibodies for 60 min in the dark under agitation. All coverslips were mounted with Dako^®^ (Agilent Technologies, Santa Clara, CA, USA, for confocal) or with Prolong^®^ (Thermo Fisher Scientific, Waltham, MA, USA, for super-resolution) on microscope slides (SuperFrost^®^ Plus, VWR, Radnor, PA, USA) and examined with a Zeiss LSM980 Airyscan microscope using a plan-Apochromat 63X NA 1.4 oil immersion objective as in [40]. For immunofluorescence on spread RBCs, cells immobilized onto PLL-coated coverslips were permeabilized with 0.5% Triton X-100 for 3 min to open the RBCs and have access to the cytoskeleton overhanging the PLL-coated RBC membrane. The next steps were undertaken as in [59] using the same antibodies and dilutions as for RBCs in suspension.

### 2.7. RBC Piezo1 and Spectrin Fluorescence Quantification

Piezo1 mean fluorescence intensity (MFI) on the whole confocal image was obtained with the Zen 3.5 software (Blue Edition, Zeiss) and divided by the total RBC surface occupancy on the image (i.e., the number of RBCs multiplied by the mean RBC area, determined with the Fiji software; Appendix A). Piezo1 total fluorescence intensity per rim and dimple regions were determined on fixed RBCs in the upper view to visualize both the rim and the dimple regions. The dimple region and the whole area were manually drawn, and the rim region area was determined as explained in Section 2.3. The mean intensity value in each area was then determined and multiplied by its respective area to obtain the total intensity in each one (Appendix A). Data were then expressed as percentages of the control condition for each area. To determine the spectrin membrane occupancy, we used the Fiji software on isolated RBCs with smooth surface. RBCs were surrounded and an arbitrary threshold was applied and maintained constant for all analyses to discriminate between the pixels that correspond to spectrin labeling (quantified in the analysis) and the ones that are from the background (not quantified). The percentage of occupied area by the spectrin pixels was then expressed by reference to the control condition (Appendix A).

### 2.8. RBC Membrane Cholesterol Content

Membrane chol content was determined using the Amplex Red chol assay kit (Invitrogen) in the absence of chol esterase. The chol levels were then reported to the corresponding Hb content and expressed in percentages of controls.

### 2.9. RBC Ghost Preparation and Western Blotting

Washed RBCs were lysed with 5 mM hypotonic PBS, and ghosts were generated by recircularization of membranes in 20 mM PBS, as in [60]. Then, an equal volume of RBC ghosts was mixed with a buffer containing 10 mM dithiothreitol (DTT) and boiled for 5 min. Each sample was then resolved by sodium dodecylsulfate polyacrylamide gel electrophoresis (Mini-Protean TGX Precast Gels 4–15% (*w*/*v*) SDS-PAGE; BioRad, Hercules, CA, USA). Samples were transferred to polyvinylidene difluoride (PVDF) methanol-activated membranes and blocked for 120 min in Tris-buffered saline Tween (TBST)-5% milk. Membranes were cut in 2 submembranes between ~250 and ~130 kDa molecular weight. The upper part was incubated overnight (O/N) with the mouse monoclonal antibody to α-β-spectrin at 1:300 dilution (Sigma-Aldrich, S3396) and used as a control of charge. The lower part was either incubated O/N with the rabbit monoclonal antibody to α-adducin at 1:5000 dilution (Abcam, Cambridge, United Kingdom, ab40760) or with the rabbit polyclonal antibody to phospho-α-adducin at 1:500 dilution (Abcam, ab53093). The next day, the membrane was incubated for 1 h with peroxidase-conjugated goat anti-rabbit (Sigma-Aldrich, A0545) or anti-mouse (Invitrogen, G2140) antibodies. All primary and secondary antibodies were diluted in TBST-5% milk. For visualization, SuperSignalTM West Pico (34580, Thermo Fisher Scientific) was used.

### 2.10. Force–Distance Curve-Based Atomic Force Microscopy

AFM experiments were performed with a Bioscope Resolve AFM (Bioscope Resolve, Bruker, Billerica, MA, USA) operated in PeakForce QNM mode at RT in DMEM. Force–distance (FD)-based multiparametric maps were acquired using a force setpoint of 300 pN, and for experiments at different indentation rates, individual FD curves were recorded in contact mode on the RBC surface, as in [61].

### 2.11. RBC Surface Phosphatidylserine Exposure

RBCs were incubated with Annexin-V FITC (Invitrogen) in DMEM at RT for 20 min and analyzed by flow cytometry with a FACS Verse (BD Biosciences, Franklin Lakes, NJ, USA) with a medium flow rate and a total analysis of 10,000 events [62]. The FlowJo v10 software was used to determine the percent of positive cells for PS exposure in the whole RBC populations. Data were expressed as a delta or percent of each experimental condition compared with its control condition.

### 2.12. Data Presentation and Statistical Analyses

Data are expressed as means ± SEM when the number of independent experiments was ≥3 or as means ± SD if *n* ≤ 2. Statistical analyses were applied only if at least 3 independent experiments were available. For 2 or more conditions compared with the control one, the Kruskal–Wallis test, followed by Dunn’s multiple comparisons test, was used for unpaired and nonparametric data. To compare 1 condition to its control, the Mann–Whitney U test was used for unpaired and nonparametric data, and the *t*-test was used for unpaired and parametric data (n > 10). In the figures, a statistical indication alone means that the tested condition is compared to the general control condition; a statistical indication above a black line means that the condition is compared to its own control condition (ns, not significant; *, *p* < 0.05; **, *p* < 0.01; ***, *p* < 0.001; ****, *p* < 0.0001). Only regressions with coefficients of determination (r^2^) > 0.5 are presented on graphs.

### 2.13. Ethics

This study involving human participants was reviewed and approved by the Medical Ethics Committee of the Université catholique de Louvain, Brussels, Belgium (B403201942024 and B403201316580). All donors gave written informed consent. Four healthy adult volunteers and four patients with hereditary spherocytosis were included.

## 3. Results

### 3.1. Piezo1 Chemical Activation Increases Calcium Content, Echinocyte Proportion and Lipid Domain Abundance

We started by incubating at RT RBCs from the main donor of the study (here named C1) with increasing concentrations of Yoda1. We observed a dose-dependent increase in echinocytes at the expense of stomatocytes and discocytes, while spherocyte abundance remained unaffected (Figure 1A,C–F). RBC intracellular Ca^2+^ content increased in the same proportion (Appendix A) and positively correlated with the proportion of echinocytes and thus negatively with the proportion of stomatocytes and discocytes (Appendix A). These morphology changes were not accompanied by an increase in PS surface exposure, indicating preservation of transversal asymmetry, at least for this phospholipid (Appendix A). In contrast, a dose-dependent increase in GM1-enriched domain abundance was observed specifically in the RBC center, corresponding to the dimple region of RBCs in suspension [63] (Figure 1B,G). This increase positively correlated with echinocyte number and Ca^2+^ content and negatively with the stomatocyte and discocyte abundance (Appendix A). In addition to GM1-enriched domains, SM- and chol-enriched domains in the dimple also increased in abundance upon Yoda1 treatment (Appendix A, arrowheads), as expected from the SM-enriched domain increase together with Ca^2+^ efflux upon mechanical RBC stimulation (Appendix A) and from chol co-enrichment in GM1- and SM-enriched domains in the dimple of RBCs at resting state [47]. More surprisingly, the number of chol-enriched domains in the rim also increased (Appendix A, arrows). Altogether, these data indicated that, upon Piezo1 chemical activation by Yoda1, a Ca^2+^ influx was observed together with an increase in echinocytes and lipid domains.

### 3.2. Piezo1 Clusters Are Larger in the RBC Dimple Region and More Associated with Spectrin in the Rim Region

To then assess the distribution of Piezo1 at the RBC surface, we used Airyscan microscopy in ‘super-resolution’ (SR) mode. On spread RBCs, we observed Piezo1 clusters that could reach ~300 nm in diameter, particularly in the RBC center (Figure 2A left, inset). We also showed, using co-labeling with spectrin, that Piezo1 clusters partially associated with the spectrin meshwork especially at the RBC periphery (Figure 2A, center and right, and 2B, yellow arrowheads). This analysis was extended to RBCs in suspension to better visualize the rim and the dimple regions and confirmed the above observations by both Airyscan ‘super-resolution’ (SR) and confocal laser scanning microscopy (CLSM) (Figure 2C–F).

### 3.3. RBCs from Donors with High Echinocyte Proportion Display Higher Calcium Content, GM1-Enriched Domains and Piezo1 Fluorescence in the Dimple Region but Lower Spectrin Membrane Occupancy

To test the potential involvement of lipid domains and spectrin cytoskeleton in Piezo1 regulation as a function of RBC morphology, we compared resting RBCs from three donors (C1–C3) showing distinct proportions of stomatocytes and discocytes vs. echinocytes. We also developed methods to quantify Piezo1 by measuring its MFI on the whole RBC population and its specific association with rim or dimple regions on the whole population except spherocytes (as they no longer exhibit distinct rim and dimple regions). Such quantification indirectly reflects the Piezo1 cluster abundance, as explained in the Discussion section.

As compared to the C1 donor, C2 and C3 exhibited a decreased proportion of stomatocytes and discocytes at the benefit of echinocytes and spherocytes (Figure 3A,D–G and Appendix A; white columns). These morphological differences did not result from differential total, rim or dimple region surface areas (Appendix A; white columns) but could be linked to the lower spectrin membrane occupancy (Figure 3H; white columns) and/or the higher number of GM1-enriched domains (Figure 3B,I; white columns). Differences between donors were not due to the anticoagulant used since differential RBC morphology proportion, spectrin membrane occupancy and chol-enriched domain abundance were also seen in RBCs from tubes coated with citrate instead of EDTA/K+ (Appendix A).

Interestingly, RBCs from donors with a higher proportion of echinocytes (C2 and C3) showed higher intracellular Ca^2+^ content (Figure 3J; white columns) and Piezo1 MFI and total fluorescence in the dimple as compared to C1 (Figure 3C,K,L; white columns). In contrast, a similar Piezo1 fluorescence in the rim was observed in the three donors (Figure 3C,M; white columns).

### 3.4. Piezo1 Chemical Activation of RBCs from Donors with High Echinocyte Proportion Further Raises the Calcium Content, GM1-Enriched Domains and Piezo1 Fluorescence in the Dimple Region but Not Spectrin

In parallel, we assessed the same parameters upon Piezo1 activation by Yoda1 in RBCs from the three donors. The concentration of 100 nM was selected, except for the Piezo1 distribution, which was assessed at 50 nM Yoda1 to avoid a too high increase in echinocyte proportion resulting from the combined effect of Yoda1 and fixation (Appendix A). Whatever the donor, Yoda1 increased the echinocyte proportion at the expense of the stomatocytes and discocytes but also the abundance of GM1-enriched domains and the intracellular Ca^2+^ content (Figure 3A,B,D–G,I,J and Appendix A; orange columns). Conversely, whereas spectrin membrane occupancy increased in C1 upon Yoda1, it was not the case in C2 and C3 (Figure 3H; orange columns). Regarding Piezo1 fluorescence, the effect of Yoda1 was most evident on Piezo1 MFI and fluorescence in the dimple than in the rim, whatever the donor considered. Nevertheless, C1 systematically showed the most pronounced effect (Figure 3C,K–M; orange columns), as for spectrin membrane occupancy.

Altogether, these data showed that, upon Piezo1 chemical activation, Piezo1 clusters increased mainly in the dimple region. Moreover, the last parameter correlated with Piezo1 MFI and the intracellular Ca^2+^ content in RBCs at rest and upon Yoda1 incubation (Appendix A). We also found positive correlations between Ca^2+^ content, GM1-enriched domain abundance and echinocyte proportion (Appendix A), suggesting the interplay between GM1-enriched domains and Ca^2+^ influx through Piezo1 in relation with the RBC morphology.

### 3.5. The Calcium Content, GM1-Enriched Domains, Echinocyte Propotion and Piezo1 Fluorescence in the Dimple Region, but Not the Other Parameters, Are Increased upon Piezo1 Chemical Activation in RBCs Pre-Activated at Both RT and 37 °C

Since the above data were obtained at RT, a temperature at which the plasma membrane Ca^2+^ ATPase (PMCA) pump is less active [64,65], we then extended our experiments to RBCs pre-incubated at 37 °C before Piezo1 chemical activation. The intracellular Ca^2+^ was still increased upon Yoda1 treatment of RBCs pre-incubated at 37 °C but to a lower extent than at RT (Figure 4A; orange columns). This lower increase could be explained by the optimal working temperature of the PMCA pump [64,65] but also potentially by Piezo1 pre-activation at 37 °C, as supported by the stronger intracellular Ca^2+^ level in basal conditions at 37 °C vs. RT (i.e., without Yoda1; Figure 4A; white columns) and the lower increase in Piezo1 fluorescence in the dimple region upon Yoda1 at 37 °C (Figure 4D; orange vs. white columns). Furthermore, the rise in Piezo1 MFI and fluorescence in the rim, of membrane:cytoskeleton anchorage and of chol-enriched domain abundance seen at RT were completely abrogated at 37 °C (Figure 4B,C,E,F), in contrast to the abundance of GM1-enriched domains and the proportion of echinocytes, which were even more highly increased (Figure 4G–K).

Thus, through the comparison of RT and 37 °C, we could confirm that the cytoskeleton and Piezo1 intensity in the rim behave similarly and were unaffected by Yoda1 incubation at 37 °C. The abundance of chol-enriched domains in the rim and in the dimple of RBCs followed the same trend and was unaffected upon Yoda1. In contrast to chol-enriched domains, GM1-enriched domains increased in the dimple upon Yoda1 at 37 °C, suggesting that those GM1-enriched domains could be less enriched in chol.

### 3.6. Cholesterol Depletion Impairs All the Increases Induced by Piezo1 Chemical Activation

To test the possibility that lipid domains could exhibit lower chol content, the RBC membrane was partially depleted from its chol content through the use of methyl-β-cyclodextrin (mβCD), a treatment previously validated in RBCs but also in myoblasts and breast cancer cell lines [47,66,67]. At both 0.6 and 0.9 mM, the chol content was decreased by ~25% (Figure 5A and Appendix A). In contrast, the PS surface exposure was increased at 0.9 mM but not at 0.6 mM (Appendix A); the latter was therefore chosen for subsequent experiments. As expected from a lower membrane chol content, the RBC surface area was decreased by ~15% at 0.6 mM mβCD (Appendix A) and the chol-enriched domains were completely abrogated in both the rim and the dimple regions (Figure 5B). Conversely, the number of GM1-enriched domains was preserved (Figure 5C; white columns and Appendix A), suggesting alteration of their chol content. This implies that, if the GM1-enriched domains contribute to Piezo1 activation through their appropriate composition and properties, they should no longer increase in abundance upon Yoda1. This was indeed observed (Figure 5C; orange columns). In addition to the impairment of lipid domains, a rise in stomatocytes and spherocytes at the expense of discocytes was seen (Figure 5D–G and Appendix A; white columns), suggesting changes in the global membrane properties as well. In contrast, morphological changes were unaffected by Yoda1 (Figure 5D–G; orange columns), as was the spectrin membrane occupancy (Figure 5H).

If lipid domains and/or global membrane properties contribute to control Piezo1 distribution and/or activation, these parameters should be altered as well upon chol depletion. In the basal condition, intracellular Ca^2+^ content and Piezo1 MFI were not affected (Figure 5I,J, white columns). However, Piezo1 intensity in the rim and in the dimple were both strongly decreased, but this could at least partly result from the ~15% decrease in the RBC surface area (Figure 5K,L and Appendix A). In addition, chol depletion prevented Yoda1 from activating Piezo1, as measured with the Fluo-4 probe (Figure 5I), and negatively affected Piezo1 intensity in the rim and dimple regions (Figure 5K–L, orange columns and Appendix A). These data suggested the involvement of the membrane chol, locally in domains and/or globally in the bulk membrane, in Piezo1 regulation.

### 3.7. Cholesterol Repletion Improves RBC Morphology, Chol-Enriched Domain Abundance and Piezo1 Fluorescence in the Rim Region but Not the Other Parameters

We then tested the potential reversibility of the effects through incubation of chol-depleted RBCs with mβCD:chol for 1 h and then in a chol-free medium for an additional 1 h 30 (i.e., repleted RBCs). This condition was compared to RBCs only incubated in a chol-free medium (i.e., control RBCs) and RBCs depleted and then re-incubated in a chol-free medium (i.e., depleted RBCs). Repletion allowed us to restore the initial chol content (Figure 6A) while leaving time to reform chol-enriched domains both in the rim and the dimple in an even higher number than in controls (Figure 6B). The RBC morphology was also almost completely restored (Figure 6C–F and Appendix A), but surprisingly, despite a similar chol content (Figure 6A), the global surface, the rim and the dimple areas were all increased (Appendix A), suggesting that some lipids present in the inner leaflet could have flopped to the outer leaflet during the time needed for chol repletion and re-incubation in chol-free medium. Accordingly, we measured a higher number of GM1-enriched domains vs. a lower cytoskeleton membrane occupancy in chol-repleted RBCs compared to both control and chol-depleted RBCs (Figure 6G,H and Appendix A). Regarding the alteration in the Piezo1 parameters induced by chol depletion, only Piezo1 in the rim was restored but at the same time increased as compared to controls (Figure 6I–K and Appendix A). Finally, no change in the intracellular Ca^2+^ content was detectable (Figure 6L), in agreement with the above data (see Figure 5I). All these data indicated that the strong increase in GM1-enriched domains was not restored by chol repletion. Conversely, chol repletion restored normal chol content and RBC morphology and even increased chol-enriched domain abundance and Piezo1 fluorescence in the rim region, suggesting a role for those domains in Piezo1 distribution in this area.

Testing this hypothesis should have required a co-labeling between Piezo1 and chol but was technically impossible due to lipid redistribution upon fixation and permeabilization. We therefore indirectly assessed this possibility by RBC co-labeling for Piezo1 and the chol-binding protein stomatin [24]. Stomatin was found to form very few patches at the RBC surface as previously shown [68]. Those patches were particularly visible in the rim region but also detectable in the dimple (Appendix A, red). Moreover, as hypothesized, they partially colocalized with Piezo1 in the rim region, particularly at resting state (Appendix A), supporting the differential partial association between Piezo1 clusters and lipid domains in an RBC area-dependent manner.

### 3.8. Upon Decrease in Cytoskeleton Membrane Occupancy and Stiffness in RBCs from Patients with Spherocytosis, Piezo1 Fluorescence Is Increased

To explore in more detail the mechanism behind the implication of the cytoskeleton in the control of Piezo1, we used RBCs from four splenectomized patients with hereditary spherocytosis, a disease mainly due to mutations in the genes coding for ankyrin or spectrin and characterized by a loss of RBC biconcavity and an impairment of RBC deformability [57,61,69,70]. As expected, these RBCs exhibited a decrease in ankyrin and/or spectrin contents at the benefit of the GTP-binding proteins septins and reflected by the increased septin-to-ankyrin content ratios and differential spectrin membrane occupancy (Figure 7A,B) [57]. By comparison with data obtained from different healthy donors (Figure 3), the Piezo1 MFI was increased, although non-significantly, in three out of the four spherocytosis patients (Figure 7C and Appendix A) and inversely correlated with the septin-to-ankyrin ratio and the spectrin membrane occupancy (Figure 7D,E). Thus, higher spectrin membrane occupancy was associated with lower Piezo1 MFI, supporting a restricting role of the cytoskeleton in Piezo1 distribution.

As Piezo1 mechanical sensitivity can be tuned by cell membrane tension [71], we also measured the stiffness of patient RBCs while discriminating the component due to the cytoskeleton and the one due to the membrane using AFM at high and low load, respectively. The higher the Young’s modulus, the more rigid the studied surface. Analysis at high load revealed that a higher spectrin membrane occupancy was associated with a higher cytoskeleton stiffness (Figure 7F). Analysis at low load showed that all patient RBCs exhibited a decreased membrane stiffness as compared to healthy RBCs (Figure 7G), independently from their spectrin membrane occupancy. Both cytoskeleton and membrane stiffnesses negatively correlated with Piezo1 MFI (Figure 7H,I), suggesting negative regulation of Piezo1 clustering. Nevertheless, their differential modulations in the patient RBCs and the differential slopes of the correlations suggested a differential role for cytoskeleton and membrane tension.

### 3.9. Piezo1-Independent Calcium Influx Does Not Increase GM1- and Chol-Enriched Domains, Echinocyte Proportion or Piezo1 Parameters

To finally address whether the changes in lipid domains, cytoskeleton:membrane occupancy and echinocytes could be specifically linked to the Piezo1-dependent Ca^2+^ influx, C1 and C3 were compared upon a Piezo1-independent Ca^2+^ influx through PKC activation by PMA/CalA treatment at 37 °C [56]. This treatment (i) induced the phosphorylation of adducin (Figure 8A and Appendix A), a protein involved in the stabilization of the 4.1R anchorage complex [72], (ii) decreased the extent of spectrin membrane occupancy in C1 but not in C3, which might be due to the fact that it was already very low (Figure 8B), in agreement with adducin displacement from the cytoskeleton network upon phosphorylation [73], (iii) raised the intracellular Ca^2+^ levels in both donors but at a lower extent than upon Yoda1 (Figure 8C and Appendix A), and (iv) reduced the Yoda1-induced Ca^2+^ accumulation in both donors (Appendix A). The latter observations could not be attributed to a toxic effect, as no changes in the global RBC size nor in the ratio of the dimple over the rim region area were noticed (Appendix A), but could be linked to the strong increase in the stomatocyte proportion in both donors (Figure 8D,E–H) accompanied by lower rim and dimple region area sizes (Appendix A).

In the absence of treatment, RBCs of C1 and C3 were different in terms of cytoskeleton occupancy, echinocyte proportion, intracellular Ca^2+^ content and Piezo1 parameters (Figure 8B,C,E–K; white columns), but the differences were less obvious than in Figure 3, which might be due to the pre-activation at 37 °C instead of RT. Upon PMA/CalA treatment, the echinocyte proportion was decreased to the benefit of stomatocytes (Figure 8E–H; purple columns). Piezo1 parameters and the abundance of chol-enriched domains were also lowered, therefore showing opposite effects to Yoda1 (Figure 8B,I–L; purple columns). In contrast, the abundance of GM1-enriched domains was not modified (Figure 8D,L), suggesting their specific modulation by Piezo1 chemical activation. Finally, the abundance of SM-enriched domains was increased by PMA/CalA (Figure 8D,L) as with Yoda1 treatment, confirming the entry of Ca^2+^ upon PMA/CalA and supporting the potential role of those domains in Ca^2+^ efflux. These data indicated that cytoskeleton and RBC morphology changes did not interplay with GM1- and chol-enriched domains during a Piezo1-independent Ca^2+^ influx.

## 4. Discussion

### 4.1. Data Summary

In the present study, we showed that Piezo1 submicrometric clusters were present in both the rim and the dimple regions. We also demonstrated that their proportion mainly increased in the dimple region upon Piezo1 chemical activation but also upon pre-activation at physiological temperature and in RBCs from donors with higher proportion of echinocytes. Piezo1-dependent Ca^2+^ entry involved the spectrin cytoskeleton and lipid domains in an RBC morphology-dependent manner.

### 4.2. Piezo1 Forms Submicrometric Clusters

By using CLSM and AFM, we previously revealed Piezo1 submicrometric clusters of ~275 nm in diameter at the RBC surface [40]. We could confirm here the presence of clusters by using CLSM and Airyscan microscopy in super-resolution mode. Moreover, Piezo1 MFI but also its intensity in the dimple region increased upon Piezo1 chemical activation, pre-activation by temperature increase and in donors with high echinocyte proportion. Although the abundance of clusters in either condition was not directly assessed, our data nevertheless suggested the importance of clustering in Piezo1 regulation. Indeed, we previously determined that the effect of Yoda1 on Piezo1 cluster abundance in the dimple region was ~150% vs. ~140% after quantification of fluorescence intensity in this region [40]. Likewise, in the rim region, the effect of Yoda1 on cluster abundance was ~110% vs. ~120% after fluorescence intensity quantification. As we did not observe a redistribution of the Piezo1 clusters, we assumed that the Piezo1 open conformation may lead to increased affinity of the antibody toward Piezo1 and more clusters of sufficient size to be resolved by CLSM [40].

Piezo1 clusters were also revealed or suggested in the literature. Indeed, in patch-clamp experiments of RBCs from patients with hereditary xerocytosis, groups of Piezo1 channels could show collective loss of inactivation or abrupt change in activation kinetics [38]. On human neural stem/progenitor cells (hNSPCs), Ellefsen et al. showed transient Ca^2+^ flickers upon Piezo1 activation. As they observed changes in flicker amplitude and not frequency, they assumed that this was the representation of the activity of Piezo1 clusters [74]. On HEK293T cell line stably expressing Piezo1-GFP, Ridone et al. demonstrated by super-resolution STORM microscopy in TIRF mode that Piezo1 forms nanoscale membrane clusters with a broad variety of cluster sizes [22]. By performing simulations using coarse-grained (CG) molecular dynamics (MD), another group suggested that Piezo1 channels form clusters and may positively cooperate to gate their pore [36]. On the other hand, patch-clamp on mouse neuronal N2A cells and modeling suggested moderate clustering of only two to three channels [75].

Nevertheless, a recent study on mouse RBCs using two protocols on intact and unroofed RBCs, respectively, questioned this clustering in the second protocol. The intact RBC protocol is similar to the one we used in our confocal and AFM experiments, except for the addition of a deglycosylation step before fixation and a longer time of fixation to increase antibody binding. Using this protocol, authors have shown, by 3D-SIM imaging, Piezo1 spots with a maximal diameter of 180 nm, concluding that each individual spot could feasibly contain a few Piezo1 channels tightly packed together in clusters, in agreement with our observations. In the unroofed RBC protocol, they applied a flux of medium through a needle of 20 gauge with an angle of 20° inducing the RBC opening and leaving only the poly-D-lysine-coated membrane. In those non-physiological conditions upon imaging by stimulated emission depletion microscopy (STED) and negative stain electron microscopy, they argued that the Piezo1 spots only contain one channel [76].

### 4.3. Piezo1 Clusters Increase in the Dimple Region upon Piezo1 Activation

To evaluate the distribution of Piezo1 clusters in the rim vs. the dimple region in physiological conditions, we compared RBCs from one donor (C1) at room and physiological temperature. Data revealed a strong rise in Piezo1 fluorescence in the dimple region and to a lower extent in the rim accompanied by a strong increase in intracellular Ca^2+^ content upon temperature increase, suggesting that Piezo1 pre-activation at 37 °C was accompanied by its increased association with the RBC dimple region. We also compared RBCs from C1, C2 and C3 with a huge difference in discocyte vs. stomatocyte vs. echinocyte proportion already at resting state. As we used the same RBC imaging chambers and media (i.e., same salt composition, glucose concentration and pH) for all the donors, such RBC morphological changes did not result from experimental issues shown to induce stomatocytes or echinocytes [52,77]. We therefore proposed that those differences were related to biological features modulations, in contrast to previous studies revealing that the modification of the normal famous RBC biconcave shape in vivo could only be seen as infrequent anomalies in normal blood related to no clinical diseases or pathologies [78]. Among those biological features is the membrane chol content, which is 1.3× higher in C2 and C3 than C1 [62]. Accordingly, upon chol depletion in RBCs from C1, the proportion of stomatocytes was strongly increased, in agreement with the literature [52,77]. A second factor is the differential abundance of lipid domains. As a matter of fact, C2 and C3 showed a higher number of GM1- and chol-enriched domains. Third, the membrane:cytoskeleton anchorage was also different in the three donors, showing higher spectrin membrane occupancy in C1 than in C2 and C3. Whatever the biological mechanism behind this, we showed that donors having at rest a higher abundance of echinocytes presented a higher intracellular Ca^2+^ content and a higher Piezo1 intensity in the dimple region, suggesting a higher Piezo1 basal activity. Upon Piezo1 chemical activation by Yoda1, each donor showed an increase in Piezo1 activity, as well as a preferential association in the dimple vs. rim regions.

### 4.4. Piezo1, Cytoskeleton Anchorage and Stiffness

We showed here and previously [40] on RBCs from C1 a partial spatial association between Piezo1 and spectrin at both resting state and upon Yoda1. Moreover, the interplay between Piezo1 clusters, cytoskeleton and membrane stiffness was evidenced through the use of RBCs from patients with hereditary spherocytosis. Conversely, on mouse RBCs, Piezo1 does not associate to actin or spectrin, as revealed by 2D-STED microscopy [76]. Although human and mouse RBCs do not appear to differ in membrane chol content [79,80], they present different sizes and abundance per µL of blood [81]. Thus, it remains to evaluate whether the lipid lateral organization and the membrane:cytoskeleton anchorage in these smaller RBCs can also differ and impact Piezo1.

Mechanistically, the cytoskeleton anchorage to the PM could contribute to its stiffness, which is known to result not only from membrane lipids and peripheral and TM proteins but also from cytoskeletal proteins [82,83,84]. We were able to distinguish the cytoskeletal stiffness from the membrane stiffness itself, thanks to AFM at high vs. low load of RBCs from the spherocytotic patients. All patients had impaired cytoskeleton-associated stiffness as compared to RBCs from healthy donors. Moreover, a higher stiffness was associated with a higher cytoskeleton occupancy and a lower Piezo1 MFI, suggesting that the cytoskeleton-based PM stiffness restricted the capacity of Piezo1 to cluster. Furthermore, Piezo1 has been shown to interact with the inner PM leaflet phosphatidylinositol (PI), and the depletion of PI 4,5-biphosphate (PIP_2_) and PI 4-phosphate has a negative effect on the channel amplitude of activity [25,85]. It is interesting to note that the membrane:cytoskeleton anchorage protein 4.1R possesses several FERM domains, which are binding sites for PIP_2_ and PS [86,87]. In support of this hypothesis, our unpublished data on patients with spherocytosis indicate alteration in PI contents and metabolism that could partly support the impairment of Piezo1 in this disease. It remains to be elucidated whether the membrane:cytoskeleton anchorage and related stiffness could differentially control Piezo1 in the rim and dimple regions. We already showed that Piezo1 associated more with spectrin in the rim than in the dimple region.

The role of the cytoskeleton in Piezo1 activation is supported by several studies. Piezo1 was shown to be biochemically and functionally tethered to the actin cytoskeleton via the E-cadherin/β-catenin complexes that increase the amplitude of Piezo1 activation and slow the inactivation kinetics [30]. Nevertheless, upon actin modulation, opposite effects on Piezo1 activation can be seen. On the one hand, upon actin filament disruption [31], anchorage protein knockout [88] or membrane bleb formation [89], an easier activation pressure of Piezo1 was observed. On the other hand, Piezo1 spontaneous activation can be generated by the acto-myosin contractile forces, as Piezo1 activation is inhibited by myosin 2 pharmacological inhibition [35,90]. These contradictory findings can be reconciliated as two mechanotransduction modes exist: the outside-in and the inside-out, respectively characterized by external forces that induce a mechanical work on the cell which passively responds to this stimulus and by active generation of a mechanical force by motor proteins that is undertaken by the cell [91].

### 4.5. Piezo1, Membrane Cholesterol and Biomechanical Properties

In addition to cytoskeletal proteins, membrane lipids are also known to contribute to PM stiffness and could therefore help regulate Piezo1 distribution and/or activity. In support of this possibility, we have shown a negative correlation between membrane stiffness at low load and Piezo1 MFI in RBCs from patients with spherocytosis. Moreover, the Yoda1-induced increase in Ca^2+^ content and Piezo1 MFI and fluorescence in the dimple region were abrogated by chol depletion, suggesting a role of membrane chol in Piezo1 activity. The literature data on other cell types support this possibility. First, STOML3, a member of the stomatin-like family, has been reported to regulate Piezo activity by recruiting chol and forming stiffened membranes surrounding the channel [24]. Second, chol removal by mβCD modulates Piezo1 sensitivity by increasing the minimal activation pressure [22]. Chong et al. have shown that either addition or removal of chol could lead to a reduction in Piezo1 activity upon Yoda1, suggesting that optimal chol concentrations are required [23]. In addition to chol, saturated fatty acids inhibit Piezo1 by increasing membrane stiffness while polyunsaturated ones would activate Piezo1 by increasing the inactivation time due to a decrease in the membrane stiffness [28]. In agreement, Lewis and Grandl have shown that pressure prepulses that minimize membrane tension shift overall Piezo1 sensitivity to lower pressure for activation [71].

Piezo1 activity regulated by membrane tension has also been discussed regarding its specific and huge trimeric structure. More specifically, its three extended blades made of 38 TM helices each imply that Piezo1 curves the cell membrane locally into a spherical dome. Based on this structure, a membrane dome mechanism for membrane tension sensitivity was proposed in which the dome shape provides a source of potential energy for gating when the membrane comes under tension [8]. In addition, the same group showed that, beyond the direct Piezo1 perimeter, the membrane is also curved and referred to as the Piezo1 membrane footprint [92]. The coupling of the dome and the membrane footprint amplifies Piezo1 sensitivity to tension, which is the greatest in low-tension regime. What remains to be demonstrated is the relationship between Piezo1 cluster increase and the tension in the dimple region due to high NMIIA association [44].

In addition to membrane tension, we cannot exclude the possibility that chol directly interacts with Piezo1. Indeed, chol has been shown to cross-link with several regions of Piezo1. In the Piezo1 sequence, 19 CRAC and 40 CARC amino acid motifs known to be chol-binding regions were identified. Among these, only 8 CRAC and 15 CARC were found to form significant Piezo1–chol interactions [23].

### 4.6. Piezo1 and Lipid Domains

We also showed that the implication of chol in Piezo1 function involved its clustering in distinct domains in a RBC morphology-dependent manner. Indeed, whereas both chol-enriched and GM1-enriched domains increased in abundance upon Yoda1, they were differentially impacted upon chol depletion followed by re-incubation in a chol-free medium, with different effects on Piezo1 parameters. Thus, the loss of chol-enriched domains was accompanied by the decrease in Piezo1 fluorescence in the rim region, whereas GM1-enriched domains and Piezo1 MFI and fluorescence in the dimple region were both increased. The relationship between Piezo1 and chol-enriched domains in the rim was supported by the partial colocalization between Piezo1 and the chol-binding protein stomatin. Moreover, this colocalization was higher in the rim region at rest but increased in the dimple one upon Yoda1. The relationship between Piezo1 and GM1-enriched domains in the dimple region was supported by several features. First, during RBC deformation in PDMS stretching chambers, GM1-enriched domains increase upon Ca^2+^ influx through Piezo1, whereas SM-enriched domains increase during the secondary Ca^2+^ efflux [47]. Second, upon inhibition of mechanosensitive cation channels, the GM1-enriched domain increase is abrogated [47]. Third, upon chemical activation of Piezo1, a dose-dependent increase in the GM1-enriched domains and Ca^2+^ content in RBCs from healthy donors was observed. Fourth, upon Piezo1-independent Ca^2+^ influx through PKC activation [56], the GM1-enriched domain abundance was not modified.

Three questions remain to be addressed regarding the role of lipid domains in Piezo1 function. First, our data did not allow us to determine whether GM1-enriched domains play a role in Piezo1 activation or inactivation. The second question is whether GM1- and chol-enriched domains are both able to recruit Piezo1 or not, which is challenging to assess. Nevertheless, our recent data on myoblasts show that Piezo1 clusters at the front of migrating cells colocalize with chol- and GM1-enriched domains but not with those enriched in SM [93]. Moreover, Piezo1-GFP showed a ~40% colocalization with the GM1-specific cholera toxin B subunit at the surface of HEK293T cells [22]. The third question is whether these two different types of domains play a sequential role (involving lipid exchange between domains in a dynamical way) or a differential role (based on the type of activation, i.e., chemical vs. mechanical) during RBC deformation. In support of the first possibility, GM1- and chol-enriched domains in the dimple region showed an opposite response to Yoda1 based on temperature increase, suggesting that the GM1-enriched domains formed upon Yoda1 in pre-activated RBCs contained lower chol. Likewise, it has been proposed by simulation with CG-MD that membrane insertion of Piezo1 induces redistribution of chol at the outer leaflet in proximity to Piezo1, contrasting with a spacing of phosphatidylcholine and SM [94]. In support of the second possibility, the GM1-enriched domains at the RBC surface are less ordered than those enriched in chol [46] and are associated with differential RBC areas [47].

## 5. Conclusions and Impact

Altogether, our data revealed that Piezo1 distribution in clusters and function was dependent on the spectrin cytoskeleton and lipid domains in a RBC morphology-dependent manner. The physiological relevance of this interplay has been demonstrated in RBCs from healthy donors differing by their membrane chol content and distribution, cytoskeleton anchorage and RBC morphology. Its pathological implication has been shown in hereditary spherocytosis, an RBC fragility disease due to cytoskeleton impairment. Extending this study to a higher number of healthy donors might further validate our main conclusions and could open the way to predict the level of Piezo1 activation based on the abundance of GM1-enriched domains or relative proportion of echinocytes in the blood of healthy donors. Moreover, our study opens the possibility to elucidate the deregulation of Piezo1–cytoskeleton–lipid domains interplay in patients with hereditary xerocytosis, characterized by Piezo1 mutations that induce an inherited disorder of erythrocyte dehydration [19,38], but also in Piezo1-related diseases such as beta-thalassemia and sickle cell disease. Indeed, in beta-thalassemia trait subjects, RBCs have decreased Piezo1 levels and increased contents of NMIIA but, at the same time, increased intracellular Ca^2+^ and decreased echinocytes [95,96]. In sickle cell disease, Piezo1 stimulation decreases sickle RBC deformability and increases the cell propensity to sickle upon deoxygenation and to adhere to laminin [97].

## Figures and Tables

**Figure 1 biomolecules-14-00051-f001:**
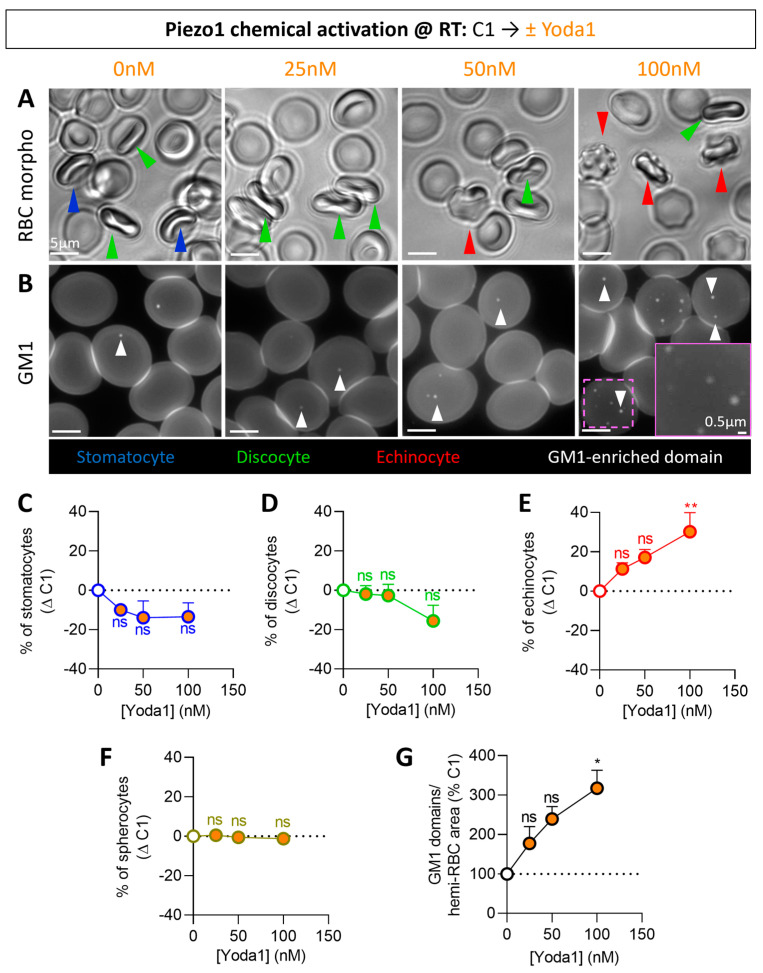
Piezo1 chemical activation by the allosteric modulator Yoda1 dose-dependently increases echinocyte proportion and GM1-enriched domain abundance. RBCs from one healthy donor (C1) were incubated in DMEM at RT for 20 min in suspension and then with Yoda1 for 30 s (white, controls; orange, Yoda1). RBCs were then analyzed for the morphology proportions (**A**,**C**–**F**) and GM1-enriched domains (**B**,**G**). (**A**,**C**–**F**) Morphology of RBCs in suspension determined in plastic IBIDI chambers. (**A**) Representative images. Stomatocytes, blue; discocytes, green; echinocytes, red; spherocytes, brown-yellow. (**C**–**F**) Quantification of the proportions of each RBC population per total RBC number. Mean ± SEM of 4 independent experiments where 40–260 RBCs per image were counted. (**B**,**G**) Fluorescence imaging of RBCs spread on poly-L-lysine (PLL) and labeled with BODIPY-GM1. (**B**) Representative images. White arrowheads, GM1-enriched domains in the dimple region. (**G**) Quantification of domains per hemi-RBC. Mean ± SEM of 4 independent experiments where 50–130 RBCs per image were analyzed. Kruskal–Wallis with Dunn’s multiple comparison test. ns, not significant; *, *p* < 0.05; **, *p* < 0.01.

**Figure 2 biomolecules-14-00051-f002:**
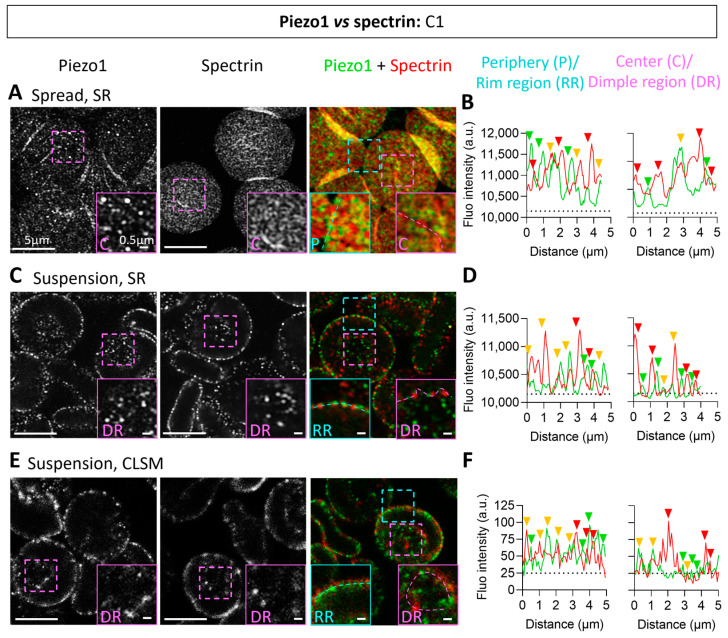
Piezo1 submicrometric clusters present different sizes and association with spectrin in the dimple and the rim regions. RBCs from one healthy donor (C1), either spread and then fixed-permeabilized (Spread) or fixed in suspension and then permeabilized (Suspension), were immunolabeled and analyzed for Piezo1, spectrin or both using super-resolution (SR, **A**–**D**) and confocal (CLSM, **E**,**F**) microscopy. (**A**,**C**,**E**) Representative images. Left, Piezo1; center, spectrin; right, Piezo1 + spectrin. Blue square, zoom of peripheral (P) or rim region (RR); pink square, zoom of center (C) or dimple region (DR). (**B**,**D**,**F**) Fluorescence intensity profiles were drawn along the rim region or the dimple region to visualize Piezo1 and spectrin. Green, Piezo1; red, spectrin; yellow, Piezo1 and spectrin association. Dotted lines, basal thresholds.

**Figure 3 biomolecules-14-00051-f003:**
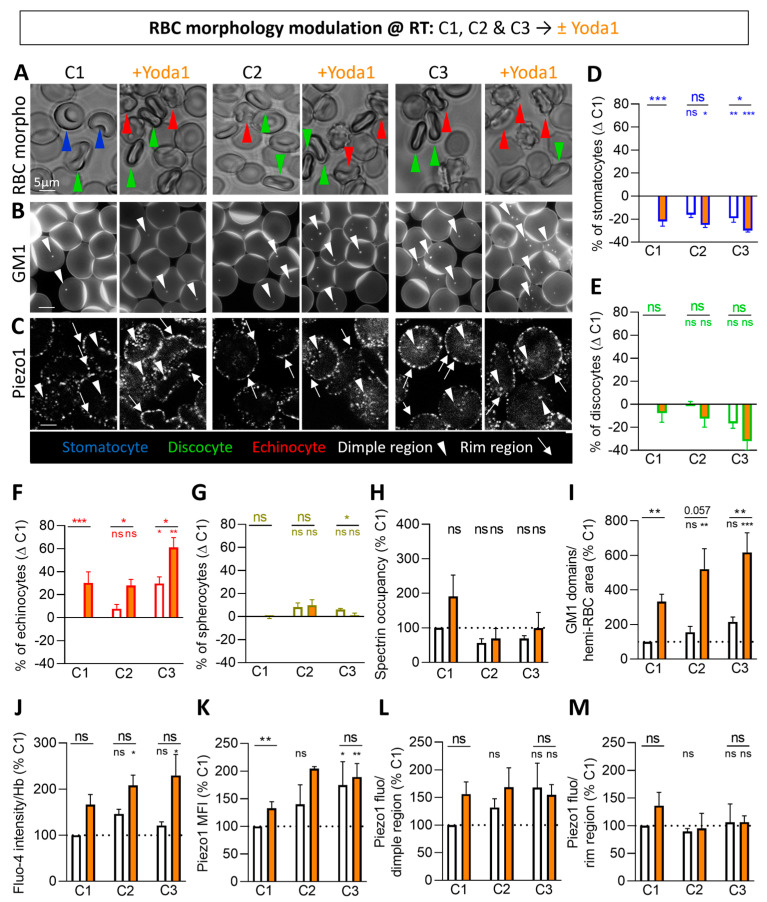
In RBCs from donors with high echinocyte proportion, GM1-enriched domains, calcium content and Piezo1 MFI and fluorescence in the dimple region are high even at resting state. RBCs from three healthy donors (C1–C3) were incubated in DMEM at RT for 20 min in suspension and then with Yoda1 for 30 s (white, controls; orange, Yoda1). RBCs were then analyzed for morphology (**A**,**D**–**G**), GM1-enriched domains (**B**,**I**), Piezo1 fluorescence (**C**,**K**–**M**), spectrin occupancy (**H**) and intracellular Ca^2+^ levels (**J**). (**A**,**D**–**G**) RBC morphology determined in plastic IDIBI chambers. (**A**) Representative images and (**D**–**G**) quantification of the proportions as in Figure 1C–F. Stomatocytes, blue; discocytes, green; echinocytes, red; spherocytes, brown-yellow. Mean ± SEM of 3–5 independent experiments where 50–250 RBCs per condition were analyzed. (**B**,**I**) Fluorescence imaging of RBCs spread on PLL and labeled with BODIPY-GM1. (**B**) Representative images and (**I**) quantification of domains as in Figure 1B,G. White arrowheads, GM1-enriched domains in the dimple region. Mean ± SEM of 4–5 independent experiments where 120–180 RBCs per image were analyzed. (**C**,**K**–**M**) Confocal imaging of RBCs fixed in suspension, spread on PLL, permeabilized and immunolabeled for Piezo1. (**C**) Representative images. White arrowheads, Piezo1 clusters in the dimple region; white arrows, Piezo1 clusters in the rim region. (**K**–**M**) Quantification of Piezo1 mean fluorescence intensity (MFI; K) and intensity in the dimple and the rim regions (**L**,**M**). Mean ± SEM of 3 independent experiments (except for C2 + Yoda1, mean ± SD of 2) where 10–12 confocal images (**K**) and 19–31 RBCs (**L**,**M**) were analyzed. (**H**) Quantification of the membrane:cytoskeleton anchorage in RBCs spread on PLL, permeabilized, fixed, immunolabeled for spectrin and imaged by confocal microscopy. Mean ± SEM of 5 independent experiments where 28–68 RBCs per condition were analyzed. (**J**) Intracellular Ca^2+^ levels determined by fluorimetry and normalized to Hb content. Mean ± SEM of 3 independent experiments. Kruskal–Wallis with Dunn’s multiple comparison and Mann–Whitney test. ns, not significant; *, *p* < 0.05; **, *p* < 0.01; ***, *p* < 0.001.

**Figure 4 biomolecules-14-00051-f004:**
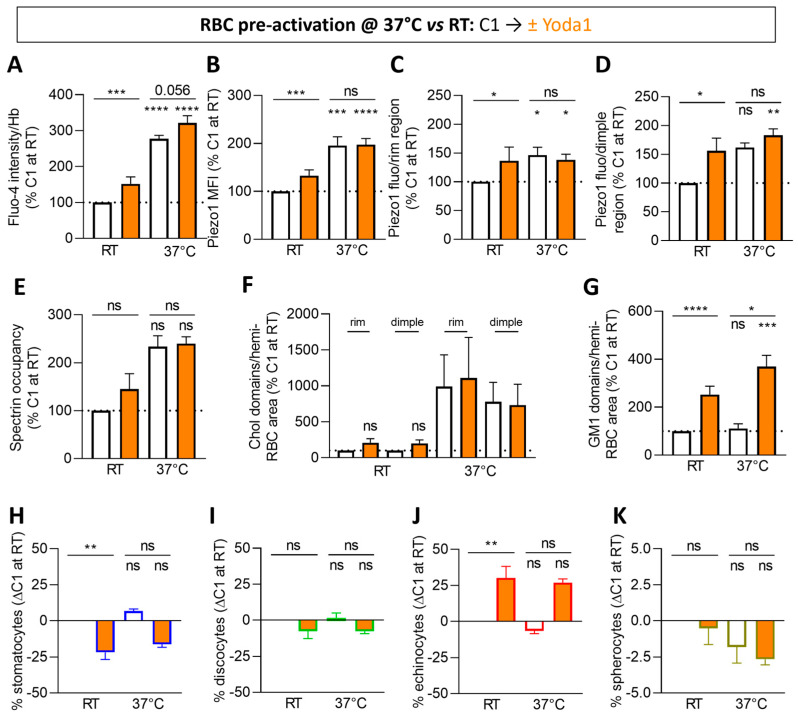
In RBCs pre-activated at 37 °C, Piezo1 MFI and fluorescence in the rim, chol-enriched domains and spectrin membrane occupancy no longer respond to Yoda1, in contrast to calcium influx, Piezo1 in the dimple, GM1-enriched domains and echinocytes. RBCs from one healthy donor (C1) were incubated in DMEM at RT or 37 °C for 20 min in suspension and then with Yoda1 for 30 s (white, controls; orange, Yoda1). RBCs were then analyzed for Ca^2+^ content (**A**), Piezo1 fluorescence (**B**–**D**), spectrin occupancy (**E**), lipid domains (**F**,**G**) and RBC morphology (**H**–**K**). (**A**) Intracellular Ca^2+^ levels were determined by fluorimetry and normalized to Hb content. Mean ± SEM of 14–31 independent experiments. (**B**–**D**) Quantification of Piezo1 MFI (**B**) and total intensity in the rim and in the dimple regions (**C**,**D**) determined as in Figure 3K–M. Mean ± SEM of 6–9 independent experiments where 10–12 confocal images (**B**) and 20–55 RBCs (**C**,**D**) were analyzed. (**E**) Quantification of spectrin membrane occupancy determined as in Figure 3H. Mean ± SEM of 3 independent experiments where 28–75 RBCs per condition were analyzed. (**F**,**G**) Quantification of chol-enriched domains in the rim and the dimple regions determined by endogenous chol decoration by the mCherry-Theta toxin fragment (**F**) and GM1-enriched domains in the dimple region determined as in Figure 1G (**G**). Mean ± SEM of 4–10 independent experiments (except for chol-enriched domains at 37 °C, mean ± SD of 2) where 60–180 RBCs per image were analyzed. (**H**–**K**) Quantification of the proportions of each RBC population per total RBC number determined as in Figure 1C–F. Mean ± SEM of 3–5 independent experiments where 50–250 RBCs per condition were analyzed. Kruskal–Wallis with Dunn’s multiple comparison and Mann–Whitney test. ns, not significant; *, *p* < 0.05; **, *p* < 0.01; ***, *p* < 0.001; ****, *p* < 0.0001.

**Figure 5 biomolecules-14-00051-f005:**
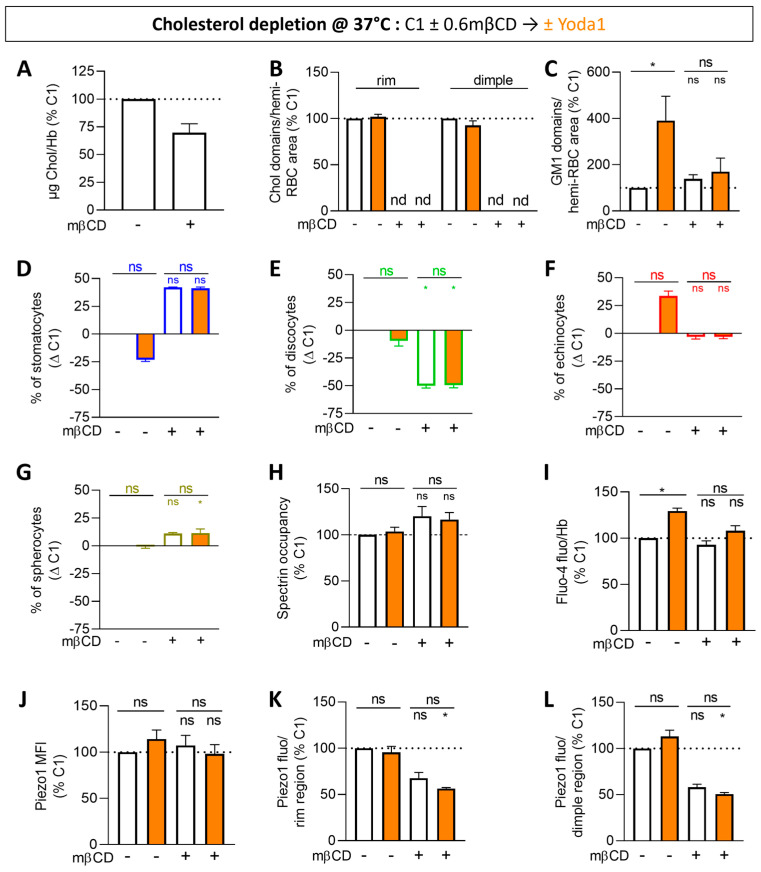
Partial cholesterol depletion abrogates chol-enriched domains and impairs the Yoda1-induced rises in GM1-enriched domains, echinocytes, calcium accumulation and Piezo1 distribution. RBCs from one healthy donor (C1) were partially depleted from chol with methyl-β-cyclodextrin (mβCD) for 15 min at 37 °C and then incubated with Yoda1 for 30 s (white, controls; orange, Yoda1). RBCs were then analyzed for the membrane chol content (**A**), lipid domains (**B**,**C**), RBC morphology (**D**–**G**), spectrin occupancy (**H**), intracellular Ca^2+^ levels (**I**) and Piezo1 fluorescence (**J**–**L**). (**A**) Membrane chol content determined by fluorimetry and normalized to Hb content. Mean ± SD of 2 independent experiments. (**B**,**C**) Quantification of chol-enriched domains in the rim and the dimple regions as in Figure 4F (**B**) and GM1-enriched domains in the dimple region determined as in Figure 1G (**C**). Mean ± SD of 2 independent experiments where 60–105 RBCs per image were analyzed (**B**). Mean ± SEM of 3–4 independent experiments where 105–180 RBCs per image were analyzed (**C**). Nd, non-detectable. (**D**–**G**) Quantification of the proportions of each RBC population per total RBC number determined as in Figure 1C–F. Mean ± SEM of 3 independent experiments where 78–202 RBCs per condition were analyzed. (**H**) Quantification of spectrin membrane occupancy determined as in Figure 3H. Mean ± SEM of 3 independent experiments where 43–75 RBCs per condition were analyzed. (**I**) Intracellular Ca^2+^ levels determined by fluorimetry and normalized to Hb content. Mean ± SEM of 3–5 independent experiments. (**J**–**L**) Quantification of Piezo1 MFI (**J**) and total intensity in the rim and the dimple regions (**K**,**L**) determined as in Figure 3K–M. Mean ± SEM of 3–6 independent experiments where 10–12 confocal images (**J**) and 23–55 RBCs (**K**,**L**) were analyzed. Kruskal–Wallis with Dunn’s multiple comparison and Mann–Whitney test. ns, not significant; *, *p* < 0.05.

**Figure 6 biomolecules-14-00051-f006:**
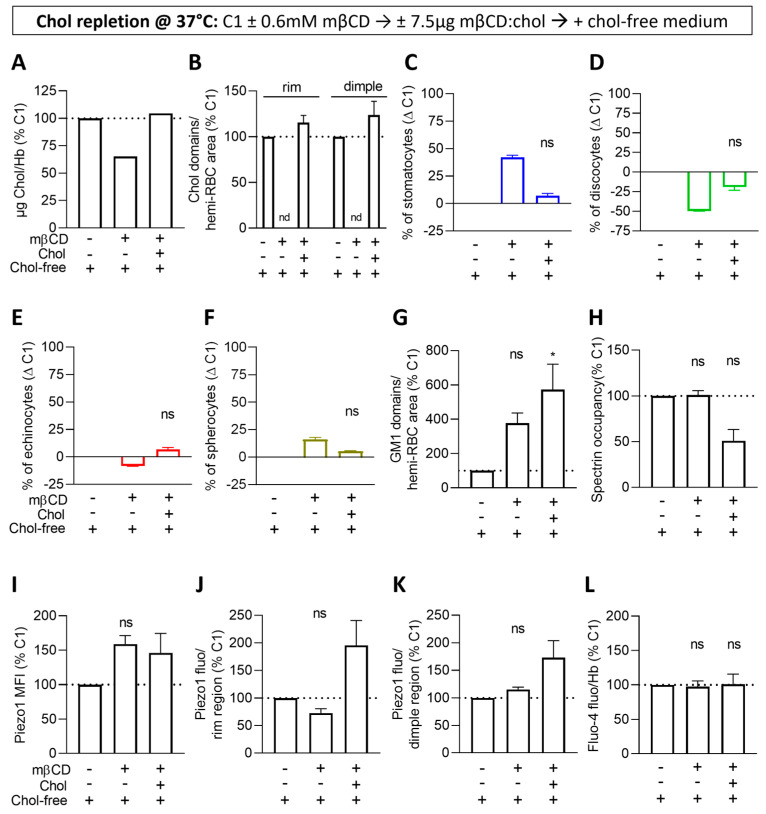
Upon cholesterol repletion, chol-enriched domains, RBC morphology and Piezo1 fluorescence in the rim are improved but not GM1-enriched domains and Piezo1 MFI and fluorescence in the dimple. RBCs from one healthy donor (C1) were either left untreated (1st column) or incubated with mβCD for 15 min followed or not (2nd column) by treatment with mβCD:chol for 60 min (3rd column). All RBCs were then re-incubated in a chol-free medium for 90 min and analyzed for the membrane chol content (**A**), lipid domains (**B**,**G**), RBC morphology (**C**–**F**), spectrin occupancy (**H**), Piezo1 fluorescence (**I**–**K**) and Ca^2+^ levels (**L**). (**A**) Membrane chol content determined by fluorimetry and normalized to Hb content. One experiment. (**B**,**G**) Quantification of chol-enriched domains in the rim and the dimple regions as in Figure 4F (**B**) and GM1-enriched domains in the dimple region as in Figure 1G (**G**). Mean ± SD of 2 independent experiments where 40–100 RBCs per image were analyzed (**B**). nd, non-detectable. Mean ± SEM of 3–4 independent experiments where 60–325 RBCs per image were analyzed (**G**). (**C**–**F**) Quantification of the proportions of each RBC population per total RBC number determined as in Figure 1C–F. Mean ± SD of 2 independent experiments (except for repletion conditions, mean ± SEM of 3) where 145–195 RBCs per condition were analyzed. (**H**) Quantification of spectrin membrane occupancy determined as in Figure 3H. Mean ± SEM of 3 independent experiments where 18–75 RBCs per condition were analyzed. (**I**–**K**) Quantification of Piezo1 MFI (**I**) and total intensity in the rim and the dimple regions (**J**,**K**) determined as in Figure 3K–M. Mean ± SEM of 3–6 independent experiments (except for control repletion condition, mean ± SD of 2) where 9–14 confocal images (**I**) and 4–24 RBCs (**J**,**K**) were analyzed. (**L**) Intracellular Ca^2+^ levels at 37 °C determined by fluorimetry and normalized to Hb content. Mean ± SEM of 3–5 independent experiments. Kruskal–Wallis with Dunn’s multiple comparison and Mann–Whitney test. ns, not significant; *, *p* < 0.05.

**Figure 7 biomolecules-14-00051-f007:**
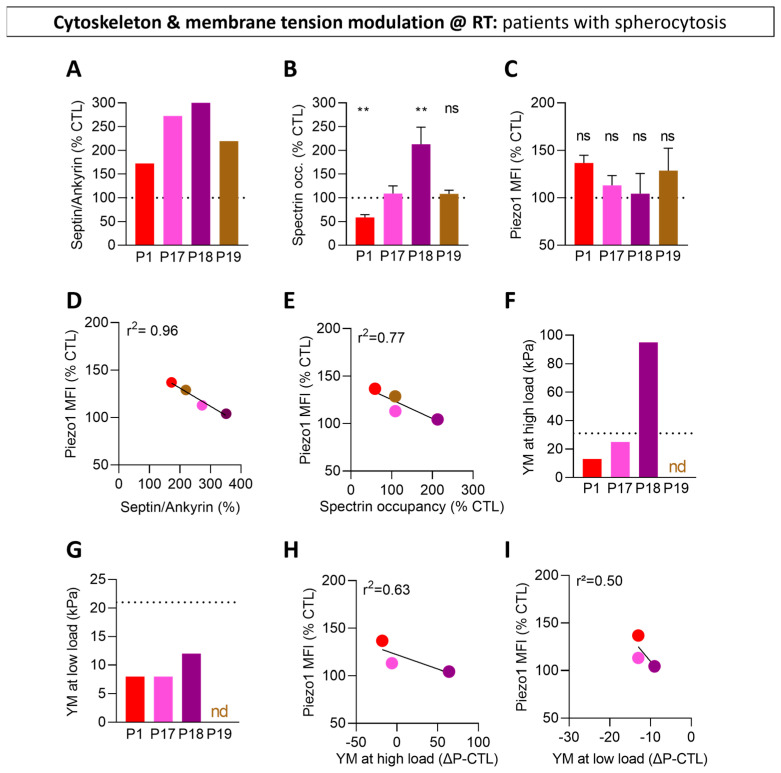
The differential impairment of RBC cytoskeleton in four patients with hereditary spherocytosis correlates with differential alterations in Piezo1 fluorescence and plasma membrane stiffness. RBCs from patients with hereditary spherocytosis (P1, red; P17, pink; P18, purple; P19, brown) were compared to sex-matched controls and analyzed for the septin/ankyrin content ratio (**A**), spectrin membrane occupancy (**B**), Piezo1 MFI (**C**) and PM stiffness (**F**,**G**). (**A**) Ratio of septin-8 over ankyrin expression. Reproduced and adapted from [57]. (**B**) Quantification of spectrin membrane occupancy. Reproduced and adapted from [57]. (**C**) Quantification of Piezo1 MFI determined as in Figure 3K. Mean ± SEM of 3 independent experiments where 4–11 confocal images per condition were analyzed. (**D**,**E**) Correlations between Piezo1 MFI and the septin/ankyrin ratio (**D**) and the spectrin occupancy (**E**). (**F**,**G**) Force–distance analysis by atomic force microscopy (AFM) at high and low loads and determination of the Young’s modulus (YM). (**H**,**I**) Correlations between Piezo1 MFI and the YM at high load (**H**) and low load (**I**). Mann–Whitney test. ns, not significant; **, *p* < 0.01.

**Figure 8 biomolecules-14-00051-f008:**
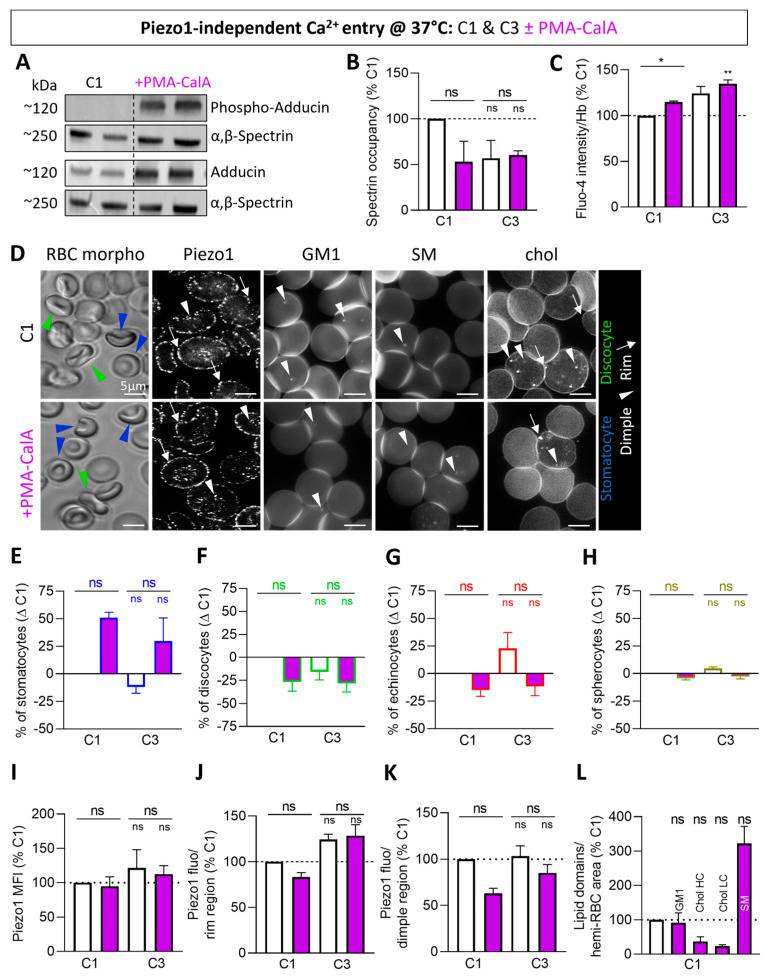
Upon Piezo1-independent calcium influx, neither echinocytes, nor Piezo1 fluorescence nor GM1- and chol-enriched domains increase. RBCs from two healthy donors (C1 and C3) were incubated at 37 °C with phorbol myristate-13-acetate (PMA) combined with calyculin A (CalA) for 20 min in suspension (white, controls; purple, PMA-CalA). RBCs were then analyzed for phospho-adducin (**A**), spectrin membrane occupancy (**B**), intracellular Ca^2+^ levels (**C**), morphology (**D**–**H**), Piezo1 fluorescence (**D**,**I**–**K**) and lipid domains (**D**,**L**). (**A**) Western Blotting of RBC ghosts analyzed in a 4–15% SDS-PAGE for expression of α-adducin, α-phospho-adducin or α-β-spectrin proteins. The spectrin expression was used as the control of charge. (**B**) Quantification of spectrin membrane occupancy determined as in Figure 3H. Mean ± SEM of 3 independent experiments where 18–46 RBCs per condition were analyzed. (**C**) Intracellular Ca^2+^ levels determined by fluorimetry and normalized to Hb content. Mean ± SEM of 3 independent experiments. (**D**–**H**) Morphology of in suspension RBCs determined in plastic IDIBI chambers. (**D**) Representative images and (**E**–**H**) quantification as in Figure 1A,C–F. Stomatocytes, blue; discocytes, green. Mean ± SEM of 3 independent experiments where 85–215 RBCs per condition were analyzed. (**D**,**I**–**K**) Confocal imaging of Piezo1 on fixed and permeabilized RBCs. (**D**) Representative images and (**I**–**K**) quantification as in Figure 3C,K–M. White arrowheads, Piezo1 clusters in the dimple region; white arrows, Piezo1 clusters in the rim region. Mean ± SEM of 3 independent experiments where 12 confocal images (**I**) and 23–55 RBCs (**J**,**K**) were analyzed. (**D**,**L**) Fluorescence imaging of RBCs labeled with BODIPY-GM1 or BODIPY-SM or with the mCherry-Theta toxin fragment for endogenous chol decoration. (**D**) Representative images and (**L**) quantification of GM1- and SM-enriched domains as in Figure 1B,G and of chol-enriched domains in the rim and the dimple regions as in Figure 4F. Mean ± SEM of 3–4 independent experiments where 75–140 RBCs per fluorescent images were analyzed. Kruskal–Wallis with Dunn’s multiple comparison and Mann–Whitney test. ns, not significant; *, *p* < 0.05; **, *p* < 0.01.

## Data Availability

Data are contained within the article and Appendix A.

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
