# Peer review of "Piezo1 Regulation Involves Lipid Domains and the Cytoskeleton and Is Favored by the Stomatocyte–Discocyte–Echinocyte Transformation"

_biomolecules, 2023, doi:10.3390/biom14010051_

Round 1

Reviewer 1 Report

Comments and Suggestions for Authors

In this nice study Stommen et al. provide a huge amount of results pointing to a regulation of mechanosensitive Piezo1 channels in native red blood cells. The authors show that Piezo1 localization in the plasma membrane can be modulated by the synthetic channel agonist Yoda1 and by temperature. They also show that depending on the membrane zone, the Ca2+ entry through Piezo1 is regulated by the spectrin cytoskeleton and by lipid domains. These results are of great interest in this rapidly emerging field. I only have a few remarks that may help improving the quality of the manuscript.

1.- I strongly recommend re-writing the results part in the abstract. Some sentences are too long and interconnected in such a way that it is difficult to follow what is being measured, what has changed and in what condition.

2.- Line 43, write “fluxes” instead of “exchanges”.

3.- Lines 55-57 use a strange wording, a hypothesis is not “characterized”. Please write “The latter proposes that the highly curved membrane imposed by Piezo1 shape contains…”

4.- Lines 64-101 provide a rather long account of several properties of RBC, but it is unclear why all those properties make these cells an ideal system to study Piezo1 regulation, as claimed in line 63

5.- In section 3.5, please, change RT by the actual recording temperature.

Comments on the Quality of English Language

I strongly recommend re-writing the results part in the abstract. Some sentences are too long and interconnected in such a way that it is difficult to follow what is being measured, what has changed and in what condition.

Author Response

Brussels, December 22nd 2023

Dear Editor,

Let us first thank you very much for your feed-back on our manuscript biomolecules-2748919, entitled ‘Piezo1 regulation involves lipid domains and the cytoskeleton and is favored by the stomatocyte-discocyte-echinocyte transformation’.

We are grateful for the analysis and the suggestions of the Reviewers. All comments were taken into serious consideration, as detailed hereafter in the point-to-point reply, and helped us to improve the manuscript.

We also revised the supplemental material. Please take this last version for publication.

We sincerely hope you will agree that all requests have been satisfactorily addressed.

Yours sincerely,

Prof. Donatienne Tyteca

DDUV Institute

UCLouvain, Brussels, Belgium

Reviewer 2 Report

Comments and Suggestions for Authors

Stommen et al., with this article, provide information regarding the mechanisms behind Piezo 1 activity. The manuscript is well-written and the authors should be praised for the study design. They performed multiple experiments to support their results and the results from one experiment led to the implementation of another experiment to answer as many questions as possible. The field of piezo-1 regulation is still obscure, and the authors shed some important light with their current findings.

Nonetheless, it would be better if the authors performed these experiments in more samples to have a real SD or SEM and not have these values based on replicates. This is my main concern for this study. The acquirement of more samples could further enhance the authors' findings and I believe that it should be mentioned as a limitation of this study since the current results should be confirmed in larger cohorts.

Apart from this, I believe that this article will be of interest to the journal's readership and a great fit to the selected Special Issue. I only have some comments that could benefit the study:

1) The authors mention a relationship between non-muscle myosin NMIIA and piezo-1. As far as I know, the entrance of calcium ions through piezo-1 can activate NMIIA, and this protein can in turn positively feedback piezo-1 activation. It was recently shown that red blood cells from beta-thalassemia trait subjects have increased levels of NMIIA, and decreased levels of piezo-1, but at the same time increased intracellular calcium. At the same time, they don't form as many echinocytes as the controls (10.3324/haematol.2020.273946; 10.3390/ijms22073369). Could these observations be partly explained by the feedback activation of piezo-1 by NMIIA, along with the current findings of the authors connecting piezo-1 with echinocytosis? If so, these erythrocytes could also be studied in terms of piezo-1 differential activity/regulation (along with hereditary spherocytosis and xerocytosis). The same could be true for sickle cell disease (10.1111/bjh.18799), therefore the authors could include more disease states in the future perspectives of this study, to broaden the impact of their results.

2) Figure S6F: Could the authors confirm that they checked if the orange dot that's quite distanced from the others is not an outlier? Especially since correlation analyses are extremely sensitive to outliers. I have the same comment for Figure 7H (purple dot).

Author Response

(The authors gave the same response as above.)
